# Analysing the impact of modifiable risk factors on cardiovascular disease mortality in Brazil

**Renato Simões Gaspar** [1] *, **Leandro F. M. Rezende** [2], **Francisco Rafael Martins Laurindo** [1]

**1** Laboratory of Vascular Biology, Health Institute (InCor), University of Sao Paulo School of Medicine, Sao Paulo, Brazil, **2** Departamento de Medicina Preventiva, Escola Paulista de Medicina, Universidade Federal de São Paulo, Sao Paulo, Brazil

* renatosgaspar@gmail.com

**Data Availability Statement:** All relevant data are within the paper and Supporting Information files.

## Abstract

### Objectives

We have examined the impact of changes in modifiable risk factors on CVD mortality in 26 Brazilian states from 2005 to 2017.

### Methods

Data were acquired from the Global Burden of Diseases study (GBD) and official sources of the Brazilian government, totalling 312 state-year observations. Population attributable fractions (PAFs) were calculated to determine the number of deaths attributed to changes in each risk factor. Fixed-effects multivariable linear regression models were performed, adjusting for income, income inequality, poverty and access to healthcare.

### Results

Between 2005 and 2017, CVD deaths reduced by 21.42%, accompanied by a decrease in smoking (-33%) and increases in hyperglycaemia (+9.5%), obesity (+31%) and dyslipidaemia (+5.2%). Reduction in smoking prevented or postponed almost 20,000 CVD deaths in this period, while increased hyperglycaemia exposure resulted in more than 6,000 CVD deaths. The association between hyperglycaemia and CVD mortality was 5 to 10 times higher than those found for other risk factors, especially in women (11; 95%CI 7 to 14, deaths per 1-point increase in hyperglycaemia exposure). Importantly, the association between hyperglycaemia and CVD mortality was independent of socioeconomic status and access to healthcare, while associations for other risk factors after the same adjustments.

### Conclusion

Reduction in smoking was the risk factor that led to the highest number of CVD deaths prevented or postponed, while hyperglycaemia showed the most deleterious association with CVD mortality. Health policies should aim to directly reduce the prevalence of hyperglycaemia to mitigate the population burden of CVD in Brazil in the future.

**Funding:** This study was funded by the São Paulo Research Foundation (FAPESP), grant 2020/15944-8 and 13/07937-8 to FRML.

**Competing interests:** The authors have declared that no competing interests exist.

## Introduction

Cardiovascular diseases (CVD) are the leading cause of death in most low- and middle-income countries (LMICs), such as Brazil [1], despite a decreasing trend in mortality over recent years [2]. Among different types of CVD, ischaemic events, such as ischaemic stroke and ischaemic heart disease have the highest impact on the Brazilian population's health [1]. The likelihood of developing CVD or dying from an ischaemic cardiovascular event is dramatically increased by the presence of common behavioural and metabolic risk factors, such as hyperglycaemia, obesity, dyslipidaemia, hypertension and smoking [3]. Among these, a recent prospective study has shown that diabetes and hypertension were the risk factors that led to the highest number of deaths due to cardiovascular events in low-income countries (LICs) [3]. However, it is unclear how different risk factors are associated with CVD mortality in LMICs, especially in countries with a universal healthcare system, given that primary care has been shown to decrease the population burden of CVD [4].

In addition to metabolic and behavioural risk factors, it has been acknowledged that the burden of CVD differs between men and women. On one hand, men display a higher mortality burden of CVD when compared to premenopausal women [5]. This trend is mostly equalized among elderly men and postmenopausal women [5]. In parallel, it was shown that women are more likely than men to present several concomitant risk factors (e.g. diabetic women tended to live more with obesity and dyslipidaemia than their male counterparts [6]). Despite this, women were less likely to be adequately treated for these conditions [6]. Altogether, it has been recognized that there are sex disparities with regard to the prevalence of risk factors and CVD mortality, however it remains yet unclear to what extent the trend of common risk factors is associated with the decrease in CVD mortality in LMICs, such as Brazil. Importantly, there may be some sort of hierarchy among risk factors in which some are more relevant than others to explain the trend in CVD at the population level.

Therefore, we have investigated the association between several common behavioural and metabolic risk factors (hyperglycaemia, obesity, dyslipidaemia, obesity and smoking) and CVD using state-level information from 2005 to 2017 in Brazil. In addition, we quantified how much of the decrease in CVD mortality between 2005 and 2017 could be explained by changes in the state-level prevalences of such risk factors. Overall, our findings may help policymakers to implement more effective strategies to target these risk factors and therefore reduce the burden of CVD in Brazil.

## Methods

### Study design and data source

We constructed panel regression models including data from 26 Brazilian states, between 2005 and 2017, totalling 312 state-year observations. Data used in this study were collected from publicly available datasets: the Global Health Data Exchange (GHDx), which is a comprehensive database on health-related variables; the Institute for Health Metrics and Evaluation (IHME), which is an independent health research centre at the University of Washington; as well as official sources from the Brazilian government, namely the Brazilian Ministry of Social Development, the Brazilian Ministry of Health and the Brazilian Institute of Geography and Statistics (IBGE). S1 Table presents details on variables and data sources used in this study.

### Cardiovascular disease events

The main dependent variables used throughout the study were the mortality or incidence rates per 100,000 individuals due to CVD, ischaemic heart disease or ischaemic stroke by sex, state

and year. A sensitivity analysis employed the years of life lost due to premature mortality (YLL, calculated for each cause-specific death relative to the normative standard life expectation at the age of death) instead of mortality rates. The full description of the International Classification of Diseases (ICD) codes contained in each dependent variable is shown in S2 Table. The quality of mortality data from Brazillian states has been attested by a previous report from the Global Burden of Disease Study 2016 (GBD 2016) that observed that all states had at least 65% of well-certified death certificates, accounting for an overall rating of 4 stars out of 5 [1].

## Behavioural and metabolic risk factors

Exposure to risk factors was computed as the summary exposure value (SEV), which was calculated as a measure of the population exposed to a certain risk factor that takes into account the extent of exposure by risk level and the severity of that risk contribution to disease burden. Risk factors included: high glucose (hyperglycaemia), high body mass index (obesity), high low-density lipoprotein (dyslipidaemia), high systolic blood pressure (hypertension) and smoking. The prevalence of type 2 diabetes mellitus was used as a sensitivity analysis to confirm the association between SEV of hyperglycaemia and mortality of cardiovascular diseases. Models that used the prevalence of diabetes were the same as those that used SEV of hyperglycaemia. The exposure definition and theoretical minimum exposure level for risk factors are presented in S3 Table [7]. All data regarding disease endpoints (mortality, prevalence, incidence or YLL) and risk factors (SEV) were age-standardized and stratified by sex.

## Covariates

Covariates were included in the model to account for socioeconomic status and access to healthcare, including the Gini index of household income, which is the most widely used metric for income inequality and was shown to be associated with several groups of NCDs in Brazil [8]; gross domestic product (GDP) in R$ (Reais, the Brazilian currency) per capita due to the known association between income and population health [9]; number of medical doctors per 1,000 and coverage of primary care [10] to account for access to healthcare; and Bolsa Família (BF, a government subsidy to low-income families) value in R$ (Reais) as a surrogate for poverty [11]. There were no missing data for observations included in the regression models. A full description of metrics used in the study is provided in S4 Table.

## Number of deaths due to changes in risk factors between 2005 to 2017

We quantified how much of the decrease in CVD events between 2005 and 2017 could be explained by changes in the state-level prevalences of behavioral and metabolic risk factors in the same period. The sex-specific population attributable fractions (PAF) of CVD events due to each risk factor (hyperglycaemia, obesity, dyslipidaemia, high systolic blood pressure and smoking) were obtained from IHME. Previous studies have established an equation to calculate the number of deaths prevented or postponed due to changes in risk factors [12, 13]. To calculate this, the number of deaths from each group of CVD in the base year of 2005 was multiplied by the difference in PAF from risk factors between the base year (2005) and final year (2017). This number was then multiplied by –1 to calculate the number of deaths that occurred due to changes in risk factors, instead of the number of deaths prevented/postponed. Therefore, a positive number indicates the number of deaths that occurred due to a given risk factor, while a negative number depicts deaths that were prevented/postponed.

## Multivariable linear regressions

Data were analysed through multivariable linear regressions with year and state fixed effects, performed using R software. Models were fitted using health outcomes for CVD (deaths per 100,000, YLLs per 100,000 or incidence per 100,000) as the dependent variable, risk factors expressed as SEV as the main independent variables and other abovementioned covariates as potential confounders. To account for heteroskedasticity and autocorrelation, which are common when dealing with panel data, we have applied clustered standard errors at the state level, as described previously [14]. Degrees of freedom, which are the values that have the freedom to vary, were calculated according to Wooldridge [14]. State fixed-effects were employed to account for unobserved cultural, geographic and historical variables that vary across states but are fixed over time. Similarly, year fixed-effects were used to account for unobserved factors that may vary throughout time but are similar in different states. Fixed-effects models are robust and suitable to analyse aggregated health data [8, 15] and were estimated using Ordinary Least Square (OLS), which is used to estimate unknown parameters in a linear regression model. The use of fixed-effects reduced the likelihood of multiplicity in the constructed model and provided a more conservative estimate. The output of the model can be interpreted as changes in the number of CVD events associated with a 1-point change in the exposure to risk factors over time.

The fully adjusted model can be written as:

$$
\begin{aligned}
CVDoutcome_{lt} &= \alpha + \beta_1 Hyperglycaemia_{lt} + \beta_2 Obesity_{lt} + \beta_3 Dyslipidaemia_{lt} + \beta_4 Hypertension_{lt} \\
&\quad + \beta_5 Smoking_{lt} + \beta_n C_{lt} + L_l + T_t + \varepsilon_{lt},
\end{aligned}
$$

where $CVDoutcome_{lt}$ is deaths per 100,000 for different CVD in state $l$ and year $t$, $Hyperglycaemia_{lt}$ is the SEV for high glucose in state $l$ and year $t$, $Obesity_{lt}$ is the SEV for high BMI in state $l$ and year $t$, $Dyslipidaemia_{lt}$ is the SEV for high LDL-cholesterol in state $l$ and year $t$, $Hypertension_{lt}$ is the SEV for high SBP in state $l$ and year $t$, $Smoking_{lt}$ is the SEV for smoking in state $l$ and year $t$ and $C_{lt}$ consists of controls lnGDP per capita, Gini Index, logBF, medical doctors per 1,000 and coverage of primary care for state $l$ and year $t$. $L$ is a state fixed-effect and $T$ is a time fixed-effect. The unadjusted model omitted control variables in $C_{lt}$, while models in which risk factors were added separately were also used. All variables included in the models were continuous.

Sensitivity analyses were performed using YLLs per 100,000 instead of deaths per 100,000. Likewise, the association between hyperglycaemia and CVD outcomes was confirmed using the prevalence of diabetes instead of the exposure to hyperglycaemia. A lag analysis was also performed, in which the mortality due to CVD was regressed with 2-year, 5-year, 8-year and 10-year lagged risk factors. Data on deaths, YLLs, prevalence and SEV for risk factors were age-standardized to correct for population ageing.

## Ethics statement

This study did not require approval from an Ethics committee since only data from publicly available secondary databases were used.

## Patient and public involvement statement

It was not appropriate or possible to involve patients or the public in the design, conduct, reporting, or dissemination plans of our research.

## Results

The age-standardized mortality and incidence of cardiovascular diseases decreased by 21% and 8%, respectively, between 2005 and 2017 in Brazil (Fig 1). In parallel, there was an increase in the SEV for hyperglycaemia (+9.5%), obesity (+31%) and dyslipidaemia (+5.2%), while the SEV for hypertension was stable (+0.001%), and that of smoking decreased by 33% in the same period. State-level data for CVD mortality and risk factors for men and women are presented in S5 Table. Moreover, descriptive information on socioeconomic and healthcare access data is presented in S6 Table.

We calculated the number of deaths that occurred due to changes in the PAF of risk factors (Table 1). Hyperglycaemia was the risk factor that accounted for the highest number of deaths, with a predicted 5119 male CVD deaths and 1254 female CVD deaths. The second-most influential risk factor was obesity, in which the best estimate predicted that there were 2981 male CVD deaths and 1109 female CVD deaths. In contrast, the drastic reduction in smoking observed throughout the years (Fig 1) led to 10039 male CVD deaths and 7267 female CVD deaths prevented or postponed. In summary, hyperglycaemia and obesity were the major causes of CVD deaths in Brazil, while the reduction in smoking led to the highest number of deaths prevented or postponed.

Multivariable linear regression analyses adjusted for GDP per capita, income inequality, poverty, access to healthcare and other risk factors were performed to examine the state-level association between the exposure to several risk factors and the mortality (Fig 2) or incidence (S1 Fig) of CVD. After including all risk factors in the model to mimic the complexity of population exposure levels and account for the effects of risk factors among themselves, only the exposure to hyperglycaemia was significantly associated with increases in CVD mortality (Fig 2), whilst there was minimal evidence of autocorrelation between hyperglycaemia and other risk factors (S2 Fig). Indeed, a 1-point increase in SEV for hyperglycaemia was associated with 11 (95%CI 7 to 14) more deaths due to CVD in women (Fig 2A). The magnitude of this association was 5 to 10 times higher than the associations reported for other risk factors. Of note, we found a negative association between SEV for high BMI and CVD mortality (Fig 2), which may be an artefact due to a strong (r = 0.92) correlation detected between SEV for high BMI and GDP per capita (S2 Fig). Despite the associations reported between hyperglycaemia and CVD mortality, we observed no association with the incidence of CVD (S1 Fig).

Considering the relevance of diabetes and hyperglycaemia to ischaemia-reperfusion injury [16], CVD were disaggregated into ischaemic diseases. Interestingly, SEV for hyperglycaemia was associated with mortality due to ischaemic stroke (2; 95%CI 1 to 3, deaths per 1 point increase in SEV) and ischaemic heart disease (3; 95%CI 2 to 5, deaths per 1 point increase in SEV) for women, while for men it was associated only with ischaemic heart disease mortality (5; 95%CI 2 to 8, deaths per 1 point increase in SEV; Fig 2B and 2C). Therefore, there may be sex-specific mechanisms regulating the interaction between hyperglycaemia and ischaemic diseases.

Associations observed in Fig 2 were confirmed by several sensitivity analyses. First, YLL was used instead of mortality rate as another proxy of the burden of CVD on population health (S3 Fig). Second, stepwise addition of variables reinforced the association between SEV of hyperglycaemia and CVD was not due to autocorrelation (S7 Table). Third, associations using the fully adjusted model were consistent with those found using an unadjusted model that did not include socioeconomic variables and access to healthcare into the model (S4 Fig). Fourth, models in which risk factors were added separately suggested the associations found between hyperglycaemia and CVD mortality were stable, while those found for hypertension and dyslipidaemia did not remain associated (S5 Fig). Finally, we replaced the predictor variable with

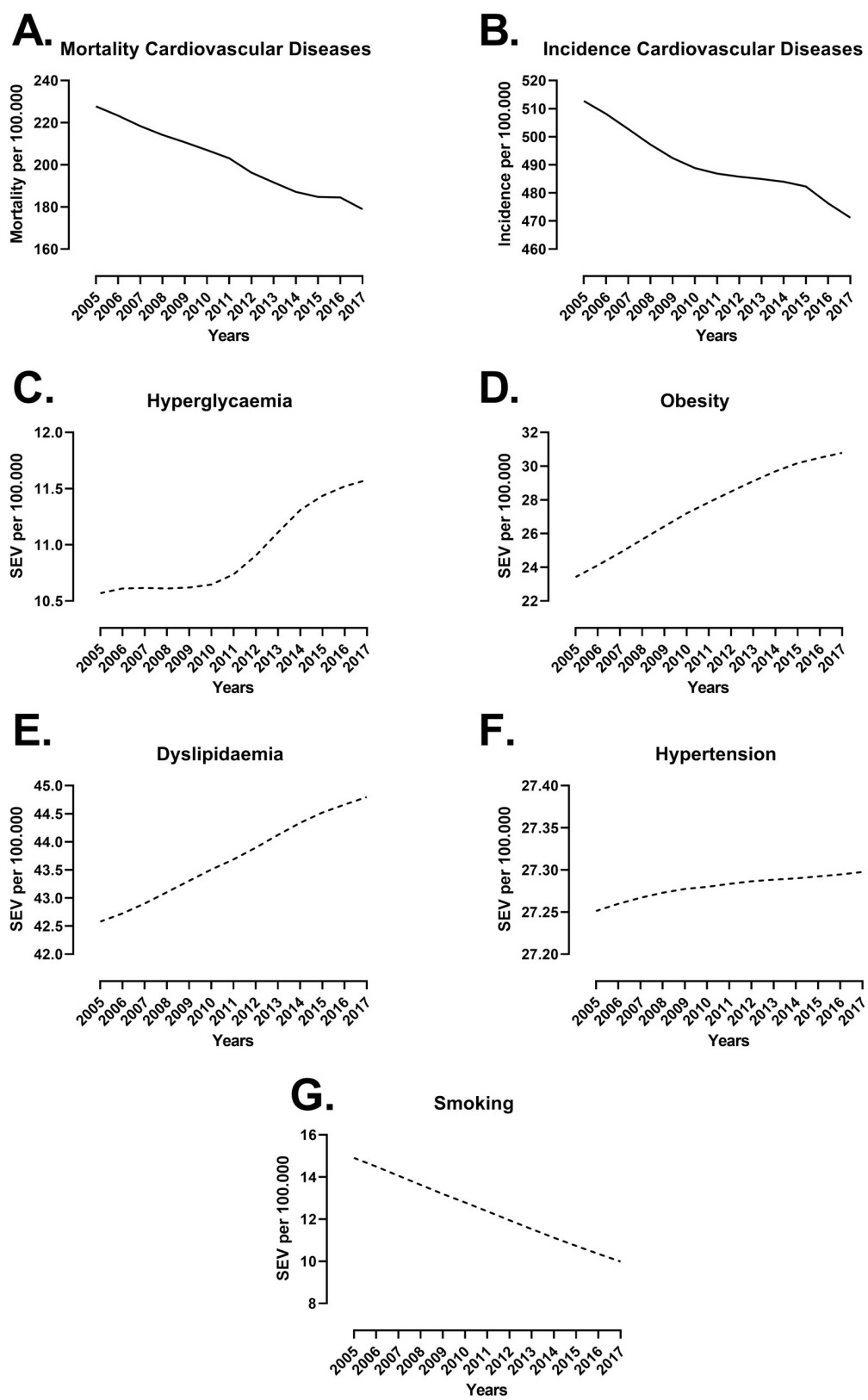

**Fig 1. Mortality and incidence of cardiovascular diseases (CVD) and summary exposure value (SEV) of risk factors in Brazil between 2005 and 2017.** Age-standardized national data for both sexes were obtained from the

Global Health Data Exchange (GHDx). Mortality (A) and incidence (B) for CVD were used as dependent variables, while SEV for hyperglycaemia (C), obesity (D), dyslipidaemia (E), hypertension (F) and smoking (G) were the main independent variables used in subsequent analyses. Data were expressed per 100,000 individuals.

the prevalence of diabetes and the associations followed the same pattern as those observed for SEV for hyperglycaemia (S8 Table). Altogether, associations reported for hyperglycaemia in Fig 2 were robust and independent of socioeconomic variables and access to healthcare. Meanwhile, associations reported for other risk factors, e.g. the negative association between BMI and CVD mortality (Fig 2), were inconsistent in the sensitivity analyses, suggesting they were due to autocorrelation or due to bias.

**Table 1. Deaths from cardiovascular diseases, ischaemic heart diseases and ischaemic stroke that were attributed to changes in risk factors in Brazilian states, 2005 to 2017.**

| Risk factors | Population attributable fraction (PAF)[a] | | | | | | Deaths attributed to changes in risk factor[b] | | |
|---|---|---|---|---|---|---|---|---|---|
| | Best (minimum to maximal) estimates | | | | | | Best (minimum to maximal) estimates | | |
| | Cardiovascular Diseases | | Ischaemic Heart Disease | | Ischaemic Stroke | | Cardiovascular Diseases | Ischaemic Heart Disease | Ischaemic Stroke |
| | 2005 | 2017 | 2005 | 2017 | 2005 | 2017 | | | |
| **SEV high glucose** | | | | | | | | | |
| *Men* | 0.19 (0.13 to 0.27) | 0.22 (0.15 to 0.32) | 0.24 (0.14 to 0.38) | 0.28 (0.16 to 0.46) | 0.26 (0.12 to 0.52) | 0.29 (0.13 to 0.59) | **5119 (3710 to 7222)** | **3231 (1816 to 5743)** | **1038 (448 to 2200)** |
| *Women* | 0.17 (0.11 to 0.25) | 0.18 (0.12 to 0.27) | 0.24 (0.13 to 0.40) | 0.26 (0.14 to 0.43) | 0.23 (0.10 to 0.50) | 0.24 (0.10 to 0.53) | **1254 (627 to 2466)** | **1039 (438 to 1975)** | **400 (64 to 756)** |
| **SEV high BMI** | | | | | | | | | |
| *Men* | 0.22 (0.14 to 0.32) | 0.24 (0.15 to 0.33) | 0.24 (0.14 to 0.34) | 0.27 (0.16 to 0.38) | 0.15 (0.08 to 0.23) | 0.16 (0.09 to 0.24) | 2981 (2849 to 2600) | 2111 (1763 to 2451) | 356 (303 to 383) |
| *Women* | 0.24 (0.16 to 0.32) | 0.24 (0.17 to 0.32) | 0.23 (0.17 to 0.33) | 0.25 (0.17 to 0.35) | 0.13 (0.08 to 0.20) | 0.14 (0.08 to 0.21) | 1109 (1532 to 262) | 1028 (1189 to 906) | 111 (33 to 114) |
| **SEV high LDL** | | | | | | | | | |
| *Men* | 0.26 (0.21 to 0.32) | 0.27 (0.22 to 0.32) | 0.50 (0.41 to 0.59) | 0.49 (0.40 to 0.59) | 0.20 (0.09 to 0.40) | 0.20 (0.08 to 0.41) | 596 (72 to 1308) | -325 (-560 to -189) | -2 (-191 to -159) |
| *Women* | 0.22 (0.17 to 0.29) | 0.23 (0.17 to 0.29) | 0.47 (0.37 to 0.58) | 0.48 (0.37 to 0.59) | 0.21 (0.06 to 0.44) | 0.21 (0.06 to 0.46) | 583 (347 to 1021) | 69 (-196 to 519) | 130 (-150 to 453) |
| **SEV high SBP** | | | | | | | | | |
| *Men* | 0.55 (0.50 to 0.60) | 0.54 (0.49 to 0.59) | 0.56 (0.48 to 0.64) | 0.55 (0.47 to 0.64) | 0.49 (0.38 to 0.60) | 0.48 (0.37 to 0.58) | -1358 (-993 to -1325) | -582 (-749 to -446) | -371 (-339 to -412) |
| *Women* | 0.54 (0.49 to 0.59) | 0.52 (0.46 to 0.59) | 0.54 (0.45 to 0.64) | 0.52 (0.42 to 0.63) | 0.47 (0.36 to 0.59) | 0.45 (0.33 to 0.58) | -2976 (-3297 to -2553) | -1108 (-1859 to -537) | -576 (-339 to -308) |
| **SEV smoking** | | | | | | | | | |
| *Men* | 0.24 (0.22 to 0.19) | 0.17 (0.22 to 0.19) | 0.31 (0.30 to 0.33) | 0.23 (0.22 to 0.25) | 0.19 (0.17 to 0.20) | 0.13 (0.12 to 0.14) | -10039 (-9658 to -10370) | -6031 (-5774 to -6281) | -1716 (-1589 to -1819) |
| *Women* | 0.17 (0.15 to 0.19) | 0.12 (0.11 to 0.19) | 0.25 (0.22 to 0.28) | 0.18 (0.16 to 0.20) | 0.12 (0.10 to 0.14) | 0.08 (0.06 to 0.09) | -7267 (-6522 to -7938) | -3834 (-3453 to -4221) | -1140 (-1589 to -1466) |
| **Number of deaths in baseline year** | | | | | | | | | |
| *Men* | | | | | | | 165,663 | 75,677 | 29,490 |
| *Women* | | | | | | | 145,925 | 57,755 | 27,828 |

[a] Calculated by the Institute for Health Metrics and Evaluation. Data are shown for the best estimates as well as the minimum and maximal values of PAF.

[b] Calculated as described in Materials and methods.

Numbers in bold indicate the highest number of deaths attributed to that risk factor. SEV: summary exposure value. BMI: body mass index. LDL: low-density lipoprotein. SBP: systolic blood pressure.

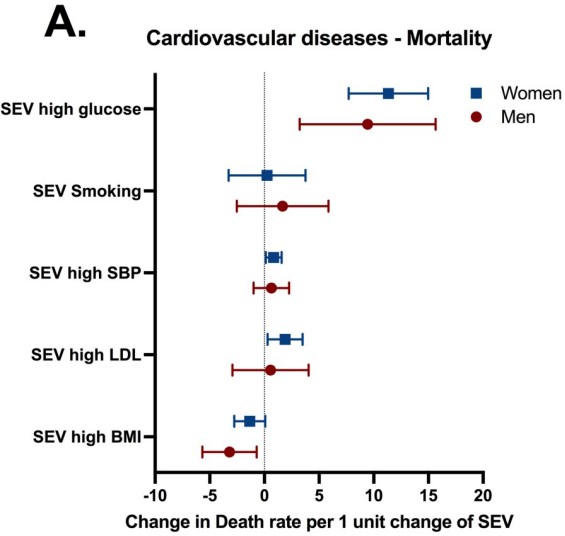

**A.**

**Cardiovascular diseases - Mortality**

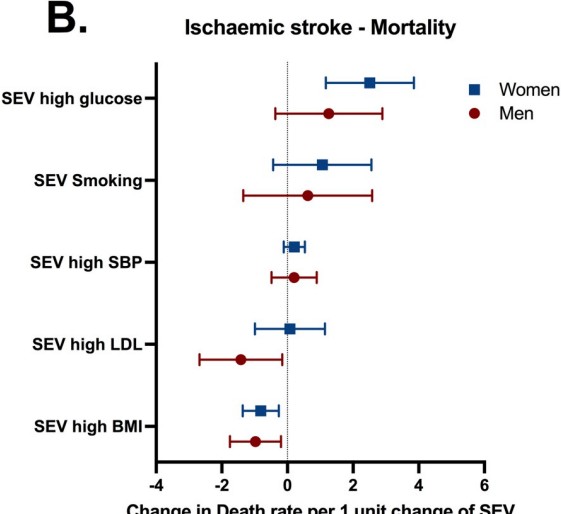

**B.**

**Ischaemic stroke - Mortality**

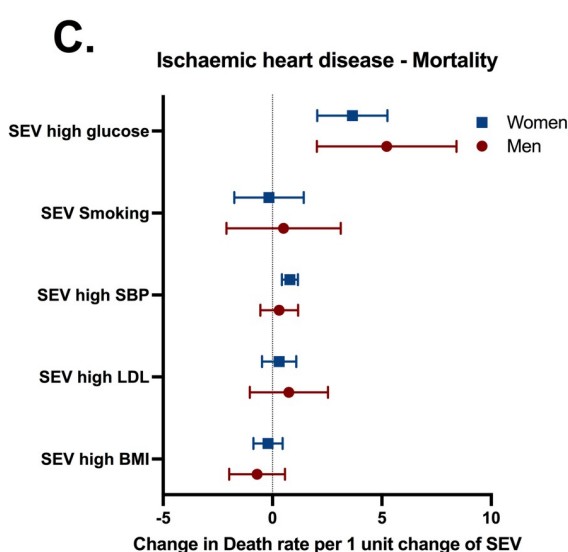

**C.**

**Ischaemic heart disease - Mortality**

**Fig 2.** Associations between SEV for risk factors and mortality rates for (A) cardiovascular diseases, (B) ischaemic stroke, and (C) ischaemic heart disease for men and women in Brazil. Variables included in the model: doctors per 1,000 habitants, hospital beds per 1,000 habitants, coverage of primary care, Bolsa Família transfer, GDP per capita, state and year fixed effects and SEV of risk factors. Mortality and SEV were per 100,000 individuals. Estimates and 95% confidence intervals (CI) are provided in the figure. SBP: systolic blood pressure. LDL: low-density lipoprotein. BMI: body mass index.

Next, we evaluated time-dependent effects of risk factors on CVD mortality (Table 2). Hyperglycaemia and hypertension were consistently associated with increases in CVD mortality rates in women. However, the magnitude of the association was 7 times higher for SEV for hyperglycaemia (7; 95%CI 3 to 11, deaths per 1 point increase in SEV for 10-year lag) than it was for SEV for hypertension (1; 95%CI 1 to 3 deaths per 1 point increase in SEV for 10-year lag). In parallel, the association between SEV for hyperglycaemia and CVD mortality in men were not observed after an 8-year lag. Finally, the lag analysis using the prevalence of diabetes instead of the SEV for hyperglycaemia (S9 Table) confirmed the data shown in Table 2. Altogether, these data collectively show a constant time-lagged association between the exposure to hyperglycaemia and CVD mortality, which seem to be more stable for women.

## Discussion

This study provides a thorough analysis of the impact of modifiable risk factors on the decreasing trend for CVD mortality in Brazil. Reductions in smoking led to the highest number of deaths prevented or postponed, while hyperglycaemia was the only risk factor with an impactful and robust association with CVD mortality, particularly for women. The association between hyperglycaemia and CVD mortality was stable after comprehensive robustness checks and was independent of socioeconomic variables and access to healthcare. Altogether, data herein gathered suggest that hyperglycaemia is associated with CVD mortality in the Brazilian

**Table 2. Lag-time analysis between the summary exposure value (SEV) for risk factors and mortality by cardiovascular diseases (CVD) in men and women, Brazil, 2005 to 2017.**

| | Men | | | | Women | | | |
|---|---|---|---|---|---|---|---|---|
| | 2-year lag (Coefficient, 95% CI) | 5-year lag (Coefficient, 95% CI) | 8-year lag (Coefficient, 95% CI) | 10-year lag (Coefficient, 95% CI) | 2-year lag (Coefficient, 95% CI) | 5-year lag (Coefficient, 95% CI) | 8-year lag (Coefficient, 95% CI) | 10-year lag (Coefficient, 95% CI) |
| **SEV high glucose** | **11 (3 to 18)**\*\* | **10 (1 to 11)**\* | **10 (1 to 19)**\* | 10 (-1 to 21) | **9 (5 to 12)**\*\*\* | **6 (3 to 10)**\*\*\* | **6 (3 to 10)**\*\*\* | **7 (3 to 11)**\*\*\* |
| **SEV high BMI** | **-8 (-14 to -2)**\*\* | **-9 (-16 to -2)**\* | **-9 (-17 to -1)**\* | -8 (-16 to 1) | **-4 (-7 to -1)**\*\* | **-4 (-6 to -1)**\*\* | **-3 (-7 to -1)**\* | -3 (-7 to 1) |
| **SEV high LDL** | -3 (-8 to 1) | -6 (-12 to 1) | **-8 (-15 to -1)**\* | -8 (-17 to 1) | -1 (-2 to 1) | -1 (-3 to 1) | **-2 (-4 to -1)**\*\* | **-3 (-5 to -1)**\*\* |
| **SEV high SBP** | -1 (-2 to 2) | -1 (-3 to 2) | -1 (-3 to 2) | -1 (-4 to 3) | **1 (1 to 2)**\* | **1 (1 to 2)**\* | **1 (1 to 2)**\* | **1 (1 to 3)**\*\* |
| **SEV Smoking** | 5 (-1 to 11) | 2 (-5 to 9) | -1 (-9 to 8) | 1 (-10 to 10) | 2 (-1 to 6) | 1 (-2 to 4) | 1 (-2 to 4) | 1 (-2 to 5) |

Numbers presented in the table are the estimates of risk factors to CVD mortality per 100,000. The full model controlled for Gini Index, GDP per capita, Bolsa Família investment, hospital beds, coverage of primary care, risk factors and state and time fixed effects. Numbers in bold depict statistical significance in which the 95% confidence interval (CI) did not include 0.

\* p<0.05

\*\* p<0.01

\*\*\* p<0.001.

SEV: summary exposure value. GDP: gross domestic product. BMI: body mass index. LDL: low-density lipoprotein. SBP: systolic blood pressure.

population, that this association is independent of healthcare access and that there are sex-specific differences. Our results also showed that reductions in smoking could partially explain the decreasing trend in CVD mortality observed in Brazil between 2005 and 2017. These findings may help policymakers to construct better policies to prevent or postpone CVD mortality.

Overall, the age-standardized CVD mortality has been decreasing in Brazil over the last decades. This could be due to socioeconomic factors, such as improved GDP per capita, educational levels and primary healthcare coverage, all of which were shown to be associated with decreased CVD mortality [3, 4, 17]. Moreover, we calculate that the marked reduction of smoking in the Brazilian population has prevented or postponed around 17000 deaths in both sexes combined. Despite the annual reduction in CVD deaths observed thus far, it is likely that, as Brazil transitions to a high-income society, this trend may slow down or even reverse, as observed for several high-income countries [18].

In light of the need to find more effective strategies to reduce CVD mortality, we have explored the associations of the population exposure to common metabolic and behavioural risk factors with CVD events. After comprehensive robustness checks it was evident that hyperglycaemia was the risk factor with the most robust and stable association with CVD mortality. In parallel, hypertension, but not hyperglycaemia, was associated with the incidence of CVD. This is in agreement with a recent cohort study that showed that the magnitude of the association of diabetes and hypertension with CVD mortality was greater than that found for other metabolic risk factors in LICs [3]. However, we have not found stable associations between hypertension and CVD mortality in the Brazilian population. It is possible that, due to a universal healthcare system and a significant increase in primary healthcare coverage, hypertension has had a lower effect on CVD mortality compared to LMICs without a universal healthcare system. Indeed, effective universal healthcare can significantly reduce deaths due to CVD [19], while a global study showed that Brazil has a high proportion of control rate for hypertension when compared to other Latin American countries and LMICs [20]. Effective hypertension control has been shown to drastically reduce the risk of developing cardiovascular events [21]. In contrast, it is still unclear if the management of diabetes with glucose-lowering drugs can reduce CVD deaths, as exemplified by studies of widely-used metformin [22] and the new drug dapagliflozin [23].

We have found some sex disparities in how hyperglycaemia is associated with CVD mortality. For instance, hyperglycaemia and diabetes were consistently associated with ischaemic stroke mortality in women but not in men. This is consistent with previous studies showing that diabetes/hyperglycaemia is a stronger risk factor for stroke in women than it is in men [24, 25]. These sex-specific differences may be due to social, psychological or biological phenomena, although underlying social determinants of sex disparities are largely understudied [26]. Biologically it has been shown that women tend to have higher rates of obesity, hypertension and dyslipidemia and are less likely to receive adequate treatment for these conditions [6]. Altogether, our data agree with previous observations that hyperglycaemia/diabetes is more strongly associated with CVD mortality in women than it is in men (this topic has been summarized by an American Heart Association statement [26]) while we found no association with the incidence of CVD.

Our study has some implications for health policy. First, data suggest that health policies should aim to directly reduce the prevalence of and exposure to hyperglycaemia in Brazil. Second, the cushion provided by universal healthcare and improvements in primary care may not suffice to mitigate the deleterious effects of hyperglycaemia on CVD mortality. These assumptions are corroborated by the high-magnitude association between hyperglycaemia and CVD mortality, which was stable even after adjusting for access to healthcare. Therefore, the increase in the prevalence of diabetes might eventually be followed by increases in CVD mortality if

different strategies are not implemented. These findings raise the need for an urgent debate on more effective policies to decrease hyperglycaemia and diabetes, such as: improvements in school nutrition, price policies to reduce the consumption of sugar-sweetened beverages and ultraprocessed food and curbing marketing and availability of these products (all of which were reviewed recently in [27, 28]). However, the debate on the need for such policies and the evidence of their impact is still insipient, especially in developing countries such as Brazil.

We acknowledge several limitations in our study. We have used estimates of risk factor exposure, which are based on observational studies by independent researchers and national surveys conducted by governmental entities. It is therefore possible that the exposure to risk factors included individuals with a pre-existing cardiovascular condition, which could potentially influence our findings. Despite the measurement error of exposures and outcomes that needs to be accounted for, data collected from IHME and the GBD study are the largest database of their kind and were consistent with a large prospective study linking modifiable risk factors and CVD mortality in 21 countries [3]. Mortality measurements might have been under-represented in Brazil, especially in poorer states [1]. However, the data used were still considered of high quality [1]. Data from IHME were smoothed by IHME [1], which could potentially mask subtle changes over time. To overcome such limitations, fixed effects for state and time were used to reduce the likelihood of unobserved changes in mortality reporting being associated with changes in income inequality [29]. Data were aggregated to states, therefore further studies are needed to test associations observed in this study at the individual level. Moreover, we are unable to make causal interpretations of models fitted, despite the use of fixed effects and relevant control variables.

Our results are supportive of a strong and stable association between hyperglycaemia and CVD mortality in Brazil, which seems to be more robust in women than in men. The association between hyperglycaemia and CVD mortality may last for up to 10 years and may be independent of access to healthcare. Indeed, changes in the population exposure to hyperglycaemia led to the highest number of deaths between 2005 and 2017 in both men and women, while the reduction in smoking led to the highest number of deaths prevented or postponed. Sex disparities reiterate that diabetes and hyperglycaemia are stronger risk factors for CVD in women than in men. Altogether, our findings provide evidence that strategies to reduce smoking were key to the reduction of CVD mortality observed in Brazil over the past decades, while there is an urgent need for policies that aim to decrease hyperglycaemia in the Brazilian population in order to mitigate the burden of CVD mortality.

## Supporting information

**S1 Table. Descriptive information on variables and data sources used across 26 Brazilian states from 2005 to 2017.**
(DOCX)

**S2 Table. International Classification of Diseases (ICD) codes and hierarchy for non-communicable diseases (NCDs) included in the analysis.**
(DOCX)

**S3 Table. Metadata of risk factors used.**
(DOCX)

**S4 Table. Glossary of metrics.**
(DOCX)

**S5 Table. Descriptive information of the summary exposure value (SEV) of risk factorsa in 2005 and 2017 in the 26 Brazilian states.**
(DOCX)

**S6 Table. Descriptive information on socioeconomic indicators and care services: GDP per capita, Gini index, bolsa família investment, hospital beds, and coverage of primary healthcare rate in 2005 and 2017 in the 26 Brazilian states.**
(DOCX)

**S7 Table. Stepwise addition of variables used in the main model.**
(DOCX)

**S8 Table. Association between the prevalence of diabetes and mortality by cardiovascular diseases in men and women, Brazil, 2005 to 2017.**
(DOCX)

**S9 Table. Lag-time analysis between the prevalence of diabetes and the mortality by cardiovascular diseases in men and women, Brazil, 2005 to 2017.**
(DOCX)

**S1 Fig.** Associations between SEV for risk factors and incidence rates for (A) cardiovascular diseases, (B) ischaemic stroke, and (C) ischaemic heart disease for men and women in Brazil. Variables included in the model: doctors per 1,000 habitants, hospital beds per 1,000 habitants, coverage of primary care, Bolsa Família transfer, GDP per capita, state and year fixed effects and SEV of risk factors. Incidence and SEV were per 100,000 people. Estimates and 95% confidence intervals (CI) are provided in the figure. SBP: systolic blood pressure. LDL: low-density lipoprotein. BMI: body mass index.
(TIFF)

**S2 Fig. Autocorrelation matrix of variables used in this study.** The Pearson r is shown in cells for correlations between individual variables. Correlations > 0.7 or <-0.7 were considered indicative of autocorrelation. GDP: growth domestic product. SEV: summary exposure value. BMI: body mass index. LDL: low-density lipoprotein. SBP: systolic blood pressure. Correlations reported for females.
(JPG)

**S3 Fig.** Associations between SEV for risk factors and years of life lost (YLL) for (A) cardiovascular diseases, (B) ischaemic stroke, and (C) ischaemic heart disease for men and women in Brazil. Variables included in the model: doctors per 1,000 habitants, hospital beds per 1,000 habitants, coverage of primary care, Bolsa Família transfer, GDP per capita, state and year fixed effects and SEV of risk factors. Mortality and SEV were per 100,000 people. Estimates and 95% confidence intervals (CI) are provided in the figure. SBP: systolic blood pressure. LDL: low-density lipoprotein. BMI: body mass index.
(TIFF)

**S4 Fig.** Unadjusted associations between SEV for risk factors and mortality due to (A) cardiovascular diseases, (B) ischaemic stroke, and (C) ischaemic heart disease for men and women in Brazil. Variables included in the model consisted of state and year fixed-effects and SEV of risk factors. Mortality and SEV were per 100,000 people. Estimates and 95% confidence intervals (CI) are provided in the figure. SBP: systolic blood pressure. LDL: low-density lipoprotein. BMI: body mass index.
(TIFF)

**S5 Fig. Associations between SEV for risk factors and mortality due to cardiovascular diseases (CVD) in independent models.** Risk factors were regressed with CVD mortality in different, independent models. The unadjusted model (A) consisted of state and year fixed-effects and SEV of risk factors. The adjusted model included: doctors per 1,000 habitants, hospital beds per 1,000 habitants, coverage of primary care, Bolsa Família transfer, GDP per capita, state and year fixed effects and SEV of risk factors. Mortality and SEV were per 100,000 people. Estimates and 95% confidence intervals (CI) are provided in the figure. SBP: systolic blood pressure. LDL: low-density lipoprotein. BMI: body mass index.
(TIFF)

**S1 File.**
(XLSX)

# Acknowledgments

Authors are thankful to the research staff of the Vascular Biology Laboratory at the Heart Institute of the University of Sao Paulo.

# Author Contributions

**Conceptualization:** Renato Simões Gaspar.

**Data curation:** Renato Simões Gaspar.

**Formal analysis:** Renato Simões Gaspar.

**Funding acquisition:** Francisco Rafael Martins Laurindo.

**Methodology:** Renato Simões Gaspar, Leandro F. M. Rezende.

**Project administration:** Francisco Rafael Martins Laurindo.

**Supervision:** Leandro F. M. Rezende.

**Writing – original draft:** Renato Simões Gaspar.

**Writing – review & editing:** Renato Simões Gaspar, Leandro F. M. Rezende, Francisco Rafael Martins Laurindo.

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
