## [Decision Letter · Decision Letter 0]

20 Apr 2022

PONE-D-22-05054Analysing the impact of modifiable risk factors on cardiovascular diseases mortality in BrazilPLOS ONE

Dear Dr. Gaspar,

Thank you for submitting your manuscript to PLOS ONE. After careful consideration, we feel that it has merit but does not fully meet PLOS ONE’s publication criteria as it currently stands. Therefore, we invite you to submit a revised version of the manuscript that addresses the points raised during the review process.

We look forward to receiving your revised manuscript.

Kind regards,

Venkata Naga Srikanth Garikipati, PhD

Academic Editor

PLOS ONE

Journal Requirements:

Reviewers' comments:

Reviewer's Responses to Questions

**Comments to the Author**

1. Is the manuscript technically sound, and do the data support the conclusions?

Reviewer #1: Yes

Reviewer #2: Yes

2. Has the statistical analysis been performed appropriately and rigorously? 

Reviewer #1: Yes

Reviewer #2: Yes

3. Have the authors made all data underlying the findings in their manuscript fully available?

Reviewer #1: Yes

Reviewer #2: Yes

4. Is the manuscript presented in an intelligible fashion and written in standard English?

Reviewer #1: Yes

Reviewer #2: Yes

5. Review Comments to the Author

Reviewer #1: Authors have examined the impact of changes in modifiable risk factors on CVD mortality in 26 Brazilian states from 2005 to 2017. As the results, authors have found Reduction in smoking was the risk factor that led to the prevention of highest number of CVD deaths, while hyperglycemia showed the most deleterious

association with CVD death. Manuscript is well written and results are very important.

However, It would be interesting to know what are the cancer incidence rate among these CVD patients and their deaths and how the cancer and its therapy impact the CVD deaths.

Reviewer #2: The authors described the correlation between hyperglycaemia and CVD with disparities among different population gender groups in Brazil. It is interesting to note that women are more prone to diabetes and hyperglycaemia induced CVD than men. The study “Analysing the impact of modifiable risk factors on cardiovascular diseases mortality in Brazil" is a lucid read and the data presented through masses of statistics in the current research sets tone for the future directives with regard to CVD management in population in Brazil.

There are certain typographical errors in the writing which authors should improve.

With regard to availability of the data sharing of this research, it would be better if details where and how the data can be accessed and reused would be good.

6. PLOS authors have the option to publish the peer review history of their article (what does this mean?). If published, this will include your full peer review and any attached files.

Reviewer #1: No

Reviewer #2: No

---

## [Author Response · Author response to Decision Letter 0]

3 May 2022

Reviewer #1: Authors have examined the impact of changes in modifiable risk factors on CVD mortality in 26 Brazilian states from 2005 to 2017. As the results, authors have found Reduction in smoking was the risk factor that led to the prevention of highest number of CVD deaths, while hyperglycemia showed the most deleterious

association with CVD death. Manuscript is well written and results are very important.

However, It would be interesting to know what are the cancer incidence rate among these CVD patients and their deaths and how the cancer and its therapy impact the CVD deaths.

Reply: We appreciate the reviewer's comments regarding our manuscript. After a thorough evaluation, we have concluded that your suggestion pertaining the incidence rate of cancer among CVD patients is an interesting idea that will be explored in a future study. We ask for the reviewer's understanding on this matter and thank you again for your time in evaluating our paper.

Reviewer #2: The authors described the correlation between hyperglycaemia and CVD with disparities among different population gender groups in Brazil. It is interesting to note that women are more prone to diabetes and hyperglycaemia induced CVD than men. The study “Analysing the impact of modifiable risk factors on cardiovascular diseases mortality in Brazil" is a lucid read and the data presented through masses of statistics in the current research sets tone for the future directives with regard to CVD management in population in Brazil.

There are certain typographical errors in the writing which authors should improve.

With regard to availability of the data sharing of this research, it would be better if details where and how the data can be accessed and reused would be good.

Reply: We thank the reviewer for taking the time to analyse our manuscript. After a thorough revision (both manually and using a grammar software named Grammarly), we have found several typos, as correctly pointed out by the reviewer. These typos were corrected in the revised version. Regarding data availability, we provide the URL of each database (supplementary table 1). Moreover, we are also uploading a spreadsheet containing all data (and models) used. To improve transparency, we now provide an R file with the codes used to generate the fixed-effects regression models. We appreciate the reviewer's time and hope that our changes have improved our manuscript.

---

## [Editor Report · Decision Letter 1]

24 May 2022

Analysing the impact of modifiable risk factors on cardiovascular diseases mortality in Brazil

PONE-D-22-05054R1

Dear Dr. Gaspar

We’re pleased to inform you that your manuscript has been judged scientifically suitable for publication and will be formally accepted for publication once it meets all outstanding technical requirements.

Kind regards,

Venkata Naga Srikanth Garikipati, PhD

Academic Editor

PLOS ONE
---

## [Editor Report · Acceptance letter]

2 Jun 2022

PONE-D-22-05054R1 

Analysing the impact of modifiable risk factors on cardiovascular disease mortality in Brazil 

Dear Dr. Gaspar:

I'm pleased to inform you that your manuscript has been deemed suitable for publication in PLOS ONE. Congratulations! Your manuscript is now with our production department. 

Kind regards, 

on behalf of

Dr. Venkata Naga Srikanth Garikipati 

Academic Editor

PLOS ONE